# Genetic and microenvironmental intra-tumor heterogeneity impacts colorectal cancer evolution and metastatic development

Daniel Sobral [1,2,6], Marta Martins [3,6], Shannon Kaplan [4], Mahdi Golkaram[4], Michael Salmans[4], Nafeesa Khan[4], Raakhee Vijayaraghavan[4], Sandra Casimiro [3], Afonso Fernandes[3], Paula Borralho[3], Cristina Ferreira[5], Rui Pinto [1,2], Catarina Abreu[5], Ana Lúcia Costa[5], Shile Zhang[4], Traci Pawlowski[4], Jim Godsey[4], André Mansinho [3,5], Daniela Macedo[5], Soraia Lobo-Martins[5], Pedro Filipe[5], Rui Esteves[5], João Coutinho[5], Paulo Matos Costa [5], Afonso Ramires[5], Fernando Aldeia[5], António Quintela[5], Alex So[4], Li Liu [4✉], Ana Rita Grosso [1,2✉] & Luis Costa [3,5✉]

Colorectal cancer (CRC) is a highly diverse disease, where different genomic instability pathways shape genetic clonal diversity and tumor microenvironment. Although intra-tumor heterogeneity has been characterized in primary tumors, its origin and consequences in CRC outcome is not fully understood. Therefore, we assessed intra- and inter-tumor heterogeneity of a prospective cohort of 136 CRC samples. We demonstrate that CRC diversity is forged by asynchronous forms of molecular alterations, where mutational and chromosomal instability collectively boost CRC genetic and microenvironment intra-tumor heterogeneity. We were able to depict predictor signatures of cancer-related genes that can foresee heterogeneity levels across the different tumor consensus molecular subtypes (CMS) and primary tumor location. Finally, we show that high genetic and microenvironment heterogeneity are associated with lower metastatic potential, whereas late-emerging copy number variations favor metastasis development and polyclonal seeding. This study provides an exhaustive portrait of the interplay between genetic and microenvironment intra-tumor heterogeneity across CMS subtypes, depicting molecular events with predictive value of CRC progression and metastasis development.

[1] Associate Laboratory i4HB - Institute for Health and Bioeconomy, NOVA School of Science and Technology, Universidade NOVA de Lisboa, 2829-516 Caparica, Portugal. [2] UCIBIO – Applied Molecular Biosciences Unit, Department of Life Sciences, NOVA School of Science and Technology, Universidade NOVA de Lisboa, 2829-516 Caparica, Portugal. [3] Instituto de Medicina Molecular- João Lobo Antunes, Faculdade de Medicina de Lisboa, Avenida Professor Egas Moniz, 1649-028 Lisboa, Portugal. [4] Illumina Inc., 5200 Illumina Way, San Diego, CA 92122, USA. [5] Centro Hospitalar Universitário Lisboa Norte, Hospital de Santa Maria, Lisboa, Portugal. [6] These authors contributed equally: Daniel Sobral, Marta Martins. ✉email: lliu3@illumina.com; argrosso@fct.unl.pt; lmcosta@medicina.ulisboa.pt

Colorectal cancer (CRC) is one of the leading causes of mortality and morbidity in the world, being the most incident cancer and the second most common cause of cancer-related death[1]. The development of high-throughput sequencing technologies allowed the detection of genomic, epigenomic and transcriptomic alterations associated with CRC development and evolution, highlighting its inter-tumor heterogeneity[2–4].

CRC diversity is mainly caused by three distinct pathways of genomic instability: chromosomal instability, microsatellite instability (MSI), and CpG Island Methylator phenotype (reviewed in ref. [5]). Chromosomal instability is observed in 85% of all CRCs and is usually associated with APC loss, where chromosome segregation errors during cell division lead to large copy-number variations (CNVs)[6]. MSI is driven by the inactivation of DNA repair genes, being responsible for ~15% of all CRC cases[7]. MSI encloses recurrent genetic alterations in microsatellite regions, however these tumors also present global mutational instability. Finally, the CpG Island Methylator phenotype corresponds to a global hypermethylation of the genome, where several tumor suppressor genes are switched-off[8].

Despite the distinct pathways of genomic instability, CRC heterogeneity can also be characterized by differences in the tumor microenvironment (TME). Based on genetic, phenotypic and microenvironmental features, CRC tumors can be stratified into four consensus molecular subtypes (CMS)[9]. CMS1 tumors are marked by a stronger immune infiltration, possibly driven by increased MSI and hypermutation. In opposition, the CMS2 subtype corresponds to the canonical epithelial tumor harboring high chromosomal instability in a microsatellite stable (MSS) context. The subtype CMS3 is characterized by a global metabolic dysregulation, displaying mixed genomic and epigenomic phenotypes. Finally, the mesenchymal subtype, CMS4, shows upregulation of genes involved in epithelial-to-mesenchymal transition, being associated with the worst relapse-free and overall survival (OS)[9].

Besides determining inter-tumor diversity, genomic instability forms can simultaneously fuel the acquisition of novel genetic lesions and originate new distinct cancer cells, creating a subclonal architecture that varies dynamically throughout the tumor progression[10]. This feature, termed intra-tumor heterogeneity (ITH), has been detected in almost all cancer types and affects tumor development and clinical outcome[11,12]. In some cases, ITH has been associated with drug resistance and tumor relapse[13,14]. High genetic diversity increases the probability for the emergence of subclones with higher selective advantage that can expand, escape from the primary site and form metastases in distant locations[15]. Although genetic tumor heterogeneity can play a role in tumor progression[16], how ITH influences CRC stratification and outcomes remains unclear.

In this study we analyzed the genome and transcriptome profiles of a prospective cohort of 136 CRC tissue samples, enclosing: 20 early stage II–III primary tumors with metastatic relapse (mCRC); 92 early stage II–III primary tumors that did not develop distant metastasis (nmCRC) and 12 pairs of primary-metastasis samples. We demonstrate that mutational and chromosomal instability concertedly shape genetic and microenvironmental tumor heterogeneity, where specific cancer-related genes can be used to predict clonal diversity levels in independent CRC cohorts. We then show that tumor heterogeneity can be a prognostic marker of tumor relapse, where clonal diversity driven by CNVs favors metastatic potential and multiple events of metastatic seeding. Overall, our findings show that clonal diversity can influence microenvironment heterogeneity, differing across CMS subtypes and influencing CRC evolution and metastasis progression.

## Results

**Mutational and chromosomal instability collectively and asynchronously shape inter- and intra-tumor heterogeneity of CRC.** To assess how CRC heterogeneity influences tumor progression, we produced genomic and transcriptomic profiles of 12 pairs of primary-metastasis samples and 112 primary tumors from early-stage II–III patients (of which 20 developed distant metastasis and 92 did not) followed at Hospital Santa Maria, Lisbon, Portugal (Supplementary Table 1).

The 112 early-stage tumors enclosed the CRC diversity defined by the transcriptomic-based Consensus Molecular Subtypes (CMS) (Supplementary Fig. 1a and Supplementary Data). As reported previously[9], CMS1 samples were significantly enriched in BRAF mutations (81%), while none of the CMS2 samples (and <25% in the other groups) had mutations in BRAF (Fisher's Test p value < 0.001). Unexpectedly, we did not find an enrichment of KRAS mutations in CMS3 samples, but rather a depletion of KRAS mutations in CMS1 samples (p = 0.01). We also find a mild depletion of APC mutations in CMS1 samples (p = 0.02) (Supplementary Fig. 1b). As expected, gene set enrichment analysis of CMS subtypes unveiled: CMS1 tumors associated with immune response; CMS2 with MYC and WNT activation; CMS3 with metabolic deregulation; and CMS4 enriched in stromal infiltration, TGF-Beta activation, and angiogenesis (Supplementary Fig. 1c). Consistent with the CMS canonical features, CMS1 was associated with high mutational instability (assessed through the amount of single nucleotide variants and small indels, designated as simple somatic mutations—SSMs) and high microsatellite instability (MSI-H), whereas CMS2 presented increased copy number variants (CNVs), MSS and lower number of SSMs (Supplementary Fig. 1d, e). The CMS3 and CMS4 subtypes showed a large inter-tumor diversity regarding such genomic features.

Previous studies have shown that genomic instability is a major contributor to subclonal expansion in cancer[10]. To quantify ITH we first determined the number of subclones and respective cellular prevalence inside each tumor sample with Expands[11]. We also applied the Shannon-Index, a measure that contemplates the number of distinct cellular populations and their respective estimated frequencies to quantify cell diversity[17]. As expected, CMS1 revealed the highest number of distinct genetic subclones and ITH levels, whereas CMS2 presented the lowest clonal diversity (Fig. 1a, b, Supplementary Fig. 2a and Supplementary Data). Regardless of CMS classification, ITH levels were associated with high mutation burden and MSI (Supplementary Fig. 2b, c). Notably, primary tumors located on the left side of the colon or in the rectum were less mutagenic and consequently showed lower clonal diversity (Supplementary Fig. 2d). Such findings are consistent with previous studies showing that primary tumor sidedness impacts CRC development, with higher prevalence of left-sided tumors in CMS2 samples[18]. Indeed, in our cohort 84% of the left sided primary tumors were classified as MSS and 55% belong to CMS2 subtype (Supplementary Fig. 2e, f). Moreover, analysis of 106 early-stage CRCs of the TCGA cohort, confirmed the high levels of ITH in CMS1, MSI-H and right-sided tumors (Supplementary Fig. 2g–j).

However, ITH can be also sustained by chromosomal instability, namely through large CNVs[19]. The approach applied by Expands to detect subclones discards all CNVs that do not overlap SSMs, thus potentially providing misleading estimates of ITH for tumor samples with high chromosomal instability and low mutation rate. Notably, when measuring subclone diversity based on both mutations and CNVs with PhyloWGS[20], similar levels of ITH were observed for all CMS subtypes, MSI status and primary tumor sidedness in our cohort (Fig. 1c, Supplementary Fig. 3a, b, and Supplementary Data) and TCGA cohort

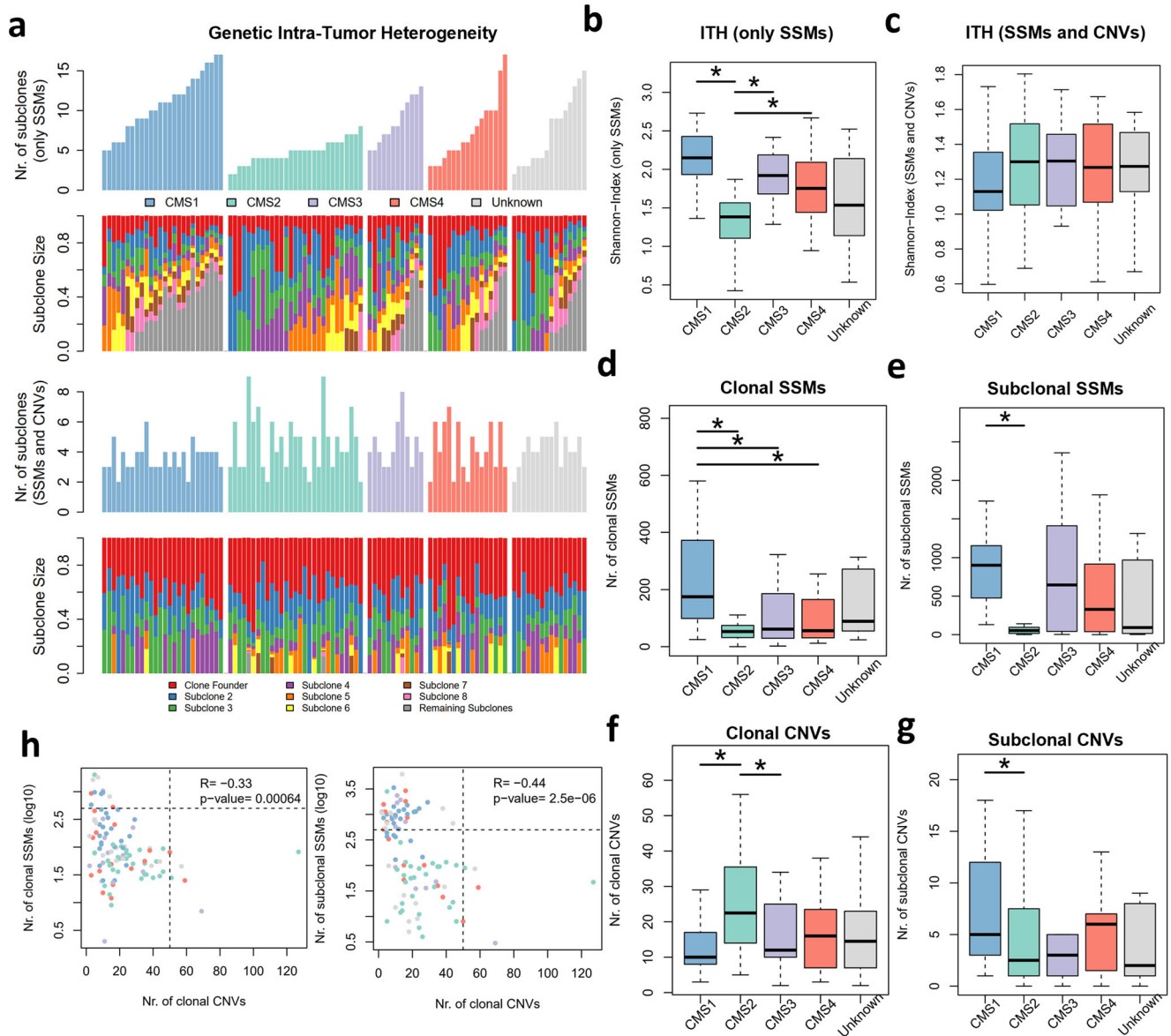

**Fig. 1 Genetic Inter- and Intra-Tumor Heterogeneity of CRC. a** Clonal composition of CRC samples grouped by CMS subtype, showing the number of subclones and respective relative frequency inferred with Expands (based on SSMs) or with PhyloWGS (using both SSMs and CNVs). **b** Distribution of the Shannon-Index values based on Expands clonal composition segregated by CMS subtype. **c** Distribution of the Shannon-Index values based on PhyloWGS clonal composition segregated by CMS subtype. **d–g** Distribution of the number of clonal SSMs (**d**), subconal SSMs (**e**), clonal CNVs (**f**), subclonal CNVs (**g**) for each CMS group. **h** Comparison between number of clonal/subclonal SSMs and clonal CNVs. Number of SSMs are represented in logarithm (log10). Estimate and statistical significance of the Pearson correlation are presented. Numbers of samples per CMS subtype: CMS1 $n = 27$; CMS2 $n = 36$; CMS3 $n = 12$; CMS4 $n = 19$; Unknown $n = 18$. *Wilcoxon signed-rank test $p$ value < 0.05.

(Supplementary Fig. 3c–e). Notably, no correlation was found between the two measures of ITH when using only SSMs or combining SSMs and CNVs in our cohort (Supplementary Fig. 3f) and TCGA cohort (Supplementary Fig. 3g).

Although similar levels of ITH were found when considering both SSMs and CNVs, the time of genomic events fueling CRC subclone diversity varied across CMS subtypes (Fig. 1d–g). CMS1 contained the highest numbers of clonal SSMs (i.e., detected in the founder clone and all subclones), confirming that early events of CMS1 tumor development are already marked by increased mutagenesis (Fig. 1d). The high number of subclonal SSMs (i.e only detected in some subclones) observed for almost all CMS subtypes (except CMS2) indicate that subsequent clonal diversity is promoted by increased mutation rates (Fig. 1e). Interestingly, larger chromosomal alterations define ITH in early events of

CMS2 development, but they appear to be more prevalent in CMS1 subsequent subclones (Fig. 1f, g). Indeed, CMS1 shows a higher proportion of subclonal SSMs and CNVs relative to CMS2 (Supplementary Fig. 4a), reinforcing CMS1 association with MSI. The clonal CNVs enriched in CMS2 affected known cancer-related genes and their respective expression levels (amplifications in *BRAF* and *PMS2*; and deletions in *TP53* and *SMAD4*, Fisher's Test $p < 0.01$, Supplementary Fig. 4b). Furthermore, CMS2 also showed higher levels of aneuploidy (Supplementary Fig. 4c), supporting that CMS2 tumor progression is driven by chromosomal instability. The differential and asynchronous preponderance of SSMs and CNVs was also observed when segregating primary tumors according to MSI classification and sidedness, confirming the parity of such stratification with CMS subtypes (Supplementary Fig. 4d, e). Such switches throughout tumor

evolution have been described, where a balance is required between the levels of genetic lesions and selective pressures[21]. Indeed, we also observed an inverse trend between the accumulation of SSMs and CNVs (Supplementary Fig. 5a). More importantly, such anti-correlation was also observed when considering only the levels of clonal SSMs and CNVs, suggesting that early excessive mutational and chromosomal instability may be incompatible in early subclonal expansion (Fig. 1h). Such balance seems to be also necessary during evolution of tumors triggered by chromosomal instability, since a negative correlation was also observed between levels of clonal CNVs and subclonal SSMs (Fig. 1h). In opposition, tumors that arise by increased mutability appear to subsequently expand by acquiring subclonal SSMs and CNVs (Supplementary Fig. 5b). This is in agreement with the perspective that early-stages of tumor evolution are marked by MSI, then followed by larger CNVs, where compensatory effects will maintain or increase the tumor fitness[21].

Thus, our findings suggest that CRC inter-tumor diversity is shaped by asynchronous and heterogenous forms of molecular alterations, where mutational and chromosomal instability concertedly boost clonal diversity in CRC.

**Distinct genetic alterations can predict intra-tumor heterogeneity across CRC diversity**. We further proceeded to identify individual genes that, when altered with SSMs or CNVs, more accurately predict CRC clonal diversity, using our previous approach based on regression modelling[22]. To reduce the noise created by passenger or silent genetic alterations, we only considered SSMs with predicted functional impact, or CNVs affecting cancer-related genes (see Methods for details). First, we modelled ITH levels assessed by Expands (i.e., based only on SSMs). The strongest predictors of high clonal diversity for all early-stage primary tumors were predominantly SSMs, where the top ranked genes were *BRAF*, *BMPR2*, *KIAA1549* and *MUC5B* (Fig. 2a and Supplementary Fig. 6a). Mutations in *BRAF* are found in about 10% of CRC patients and have long been associated with worse prognosis and resistance to standard therapies[23]. Our analysis also identified mutations in *FBXW7* tumor suppressor gene as a predictor of ITH, for which inactivating mutations were previously associated with chromosomal instability in CRC[24]. To evaluate our model, we compared the predicted values and the observed values of ITH (SSM only) for our cohort. Overall, our predictions could model 68% of the variability in CRC samples and were strongly correlated with the observed values of ITH measured by the Shannon-Index (SSM only) (Fig. 2b). More importantly, the optimal model based on our cohort could also foresee the levels of clonal diversity for the TCGA cohort, where a moderate correlation was obtained between the observed and predicted ITH levels (Fig. 2c). Since distinct genomic events can determine CRC clonal diversity, we replicated the analysis for each CMS subtype, MSI status and primary tumor location (Fig. 2a and Supplementary Fig. 6a). Our specific models could explain a high proportion of the ITH variance observed for each tumor group in our cohort, and be applied to the equivalent groups in the TCGA cohort (Fig. 2b and Supplementary Fig. 6b). The independent regression analyses identified some common predictor genes for primary tumors triggered by mutational instability (i.e., shared by CMS1, MSI-H or right-sided tumors) or chromosomal instability (i.e., CMS2, MSS or left-sided tumors) (Fig. 2a and Supplementary Fig. 6a). Thus, the predictor genes of increased ITH commonly found for highly mutated tumors were *GLI1*, *DAPK1*, *ASH1L*, *VCAM1*, *SFRP4*, *POLE*, *KRIT1*, *NRG1* (Fig. 2a). *GLI1*, *SFRP4* and *POLE* genes were previously involved in tumorigenesis with a major contribution to tumor genomic instability[25]. Curiously, the *SETD2* gene identified as predictor for

CMS1 tumors (Supplementary Fig. 6a), was recently identified as a driver of ITH in kidney cancer[22]. In opposition, the ITH levels in tumors characterized by high chromosomal instability could be modelled by *KMT2C, MAP2, SF3B1, VCAN, TP53, TCL1A and COL1A1. KMT2C* is a reported epigenetic regulator involved in the expression of DNA damage response and DNA repair genes, particularly homologous recombination-mediated DNA repair, whose downregulation leads to increased DNA damage and chromosomal instability[26]. Furthermore, mutations on the *SF3B1* gene were found as subclonal drivers in chronic lymphocytic leukemia[27]. Although most genes showed small genetic alterations, some CNVs were also selected as predictors of ITH, namely the *HOXA10* for CMS3 and right-sided tumors (Fig. 2a and Supplementary Fig. 6a).

Next, we repeated the approach to model ITH enclosing SSMs and CNVs as estimated by PhyloWGS. We could identify predictors of clonal diversity for almost all samples and also when segregating by the different tumor groups, except for CMS1 and CMS2 (Fig. 2c and Supplementary Fig. 7a). Although our models could explain ITH variability (31–100%) and showed strong correlation with the observed values, they could not correctly anticipate clonal diversity for the TCGA cohort (Fig. 2d and Supplementary Fig. 7b). Nevertheless, our model revealed several common predictors between MSI-H and right-sided tumors (Fig. 2c). Mutations in *DCC* are consistently associated with low heterogeneity levels in CMS4, MSI-H and right-sided tumors. Also, some important genes of CRC development were depicted by our approach, where some mutated genes were associated with low (*APC, TP53, KRAS, FBXW7* and *TCF7L2*) or with high (*BRAF*) levels of clonal diversity (Fig. 2c and Supplementary Fig. 7a). Curiously, mutations on *APC, TP53, KRAS, BRAF* and *FBXW7* were previously associated with clonal diversity in an early-stage of CRC development[28].

Therefore, our results indicate that by combining small and large genomic alterations of specific genes, we can predict ITH levels across tumor subtypes. More importantly, our findings reinforce the divergent molecular events that influence CRC diversity and progression and the equivalence between CMS classification, MSI status and primary tumor location.

**Clonal diversity determines CRC microenvironment heterogeneity**. Besides molecular features, CRC diversity is also characterized by different TMEs, especially regarding amount and distribution of stromal and immune cells. The CMS1 and CMS4 subtypes have been referred to as having strong immune activation[9]. On the other end of the spectrum, CMS2 is described as immune silent[29]. In agreement with this, in our samples, CMS2 tumors showed a significantly lower immune score than CMS1 or CSM4 samples, with CMS3 samples having intermediate values (Supplementary Fig. 8a). Consistently, MSS and left-sided tumors also resemble an immune silent profile (Supplementary Fig. 8b, c). As expected, CMS4 presented the highest stromal content (Supplementary Fig. 8d). Furthermore, such enrichment was also observed in tumors located in the rectum, explained by the large proportion of rectal tumors belonging to CMS4 subtype (Supplementary Fig. 8e, f).

To deeply characterize the microenvironment heterogeneity of CRC, we used RNA-based deconvolution approaches based on the single-cell profiles of CRC samples produced by Lee and colleagues[30]. Such single-cell gene expression profiles identified 35 distinct cell types composing the CRC TME, enclosing: epithelial, stromal and immune cells[30]. Epithelial cells were classified according to CMS-like cell types, with the exception of CMS4, for which the main markers were stromal cells, namely

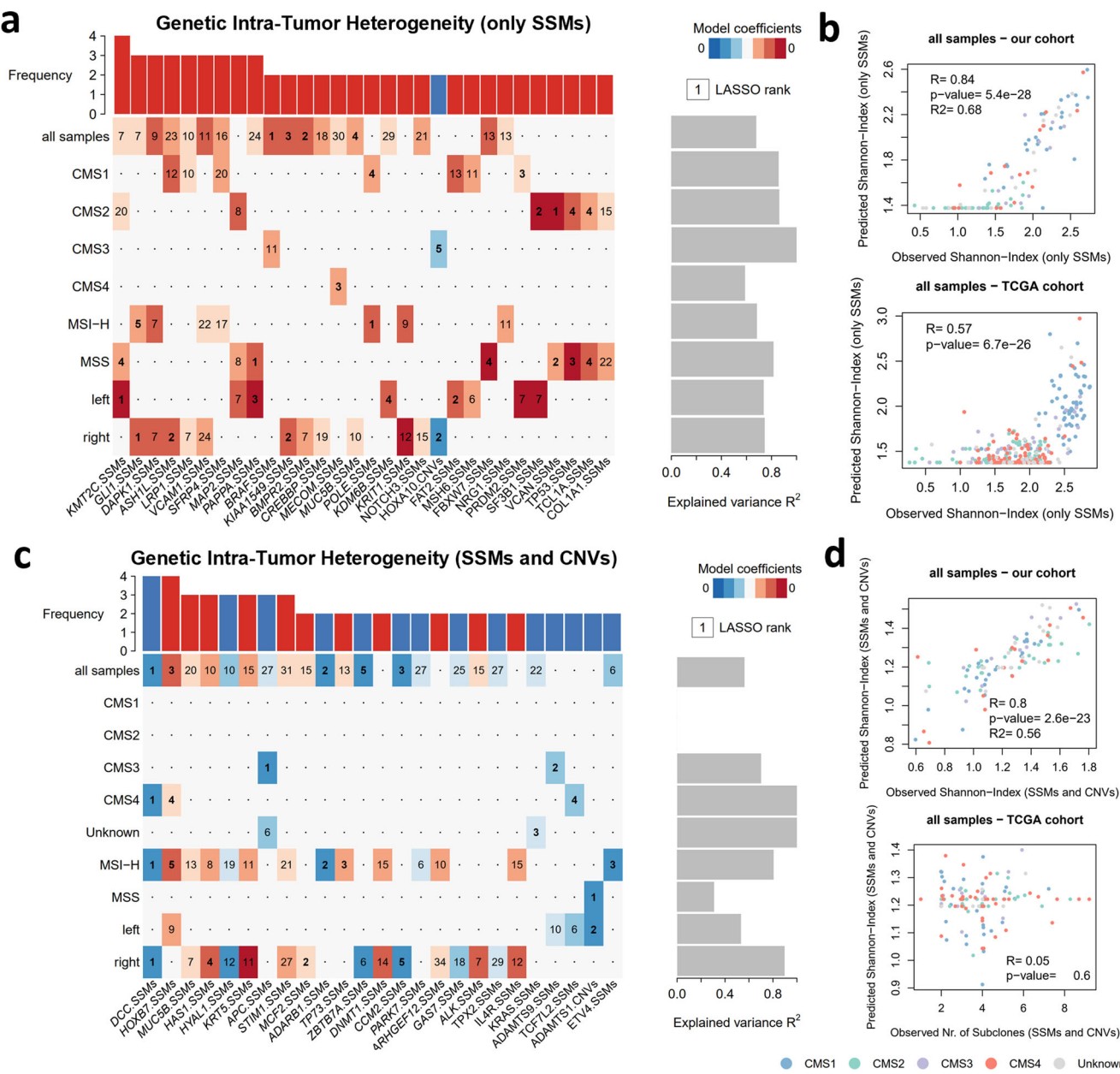

**Fig. 2 Genetic biomarker signatures for genetic ITH. a** Heatmap of cancer-related genes affected by SSMs or CNVs that are associated with genetic ITH levels (only SSMs) in CRC samples depicted by a LASSO penalized model. Each line represents an independent analysis applied to the CRC samples segregated according to CMS subtypes (all samples $n = 112$; CMS1 $n = 27$; CMS2 $n = 36$; CMS3 $n = 12$; CMS4 $n = 19$), MSI status (MSS $n = 59$; MSI-H $n = 53$) or primary tumor location (right $n = 68$; left $n = 33$). LASSO-selected coefficients are colored according to the effect of each standardized covariate in the optimal model. The numbers on each tile denote the order in which variables are included indicating their relative importance. The top bar plot indicates the frequency at which each driver-gene mutation occurs in the ITH fitted model. The right bar plot shows the explained variance. **b** Comparison between observed and predicted genetic ITH levels (only SSMs) for all samples in our cohort and in TCGA. **c** Heatmap of genes with SSMs or CNVs that are associated with genetic ITH levels (SSMs and CNVs) in CRC samples depicted by a LASSO penalized model. Graphical representation similar to (**a**). **d** Comparison between observed and predicted genetic ITH levels (SSMs and CNVs) for all samples in our cohort and in TCGA. Colors indicate CMS subtype for each CRC sample. Estimate and statistical significance of the Pearson correlation are presented. R2 represents the explained variance of the model in our cohort.

myofibroblasts[30]. Notably, multivariate analysis of the cellular composition of our cohort of primary tumors corroborated CRC diversity, where almost half of the TME variability (45.83% of variation defined by Principal Component 1) distinguish immune silent CMS2 from CMS1/CMS4 with higher immune activation (Fig. 3a). Moreover, most tumors with high MSI presented similar cellular composition, regardless of CMS classification (Supplementary Fig. 8h). According to the loadings of the principal

component analysis, CRC samples were segregated due to different cell frequencies of myofibroblasts and CMS-like signatures (Supplementary Fig. 8g). Applying a similar approach to the early-stage CRCs of the TCGA cohort confirmed the distinct TME associated with each CMS subtype (Supplementary Fig. 8i).

Lee et al. have detected that individual patients demonstrated differing intra-tumoral heterogeneity in CMS cell-types[30]. Indeed,

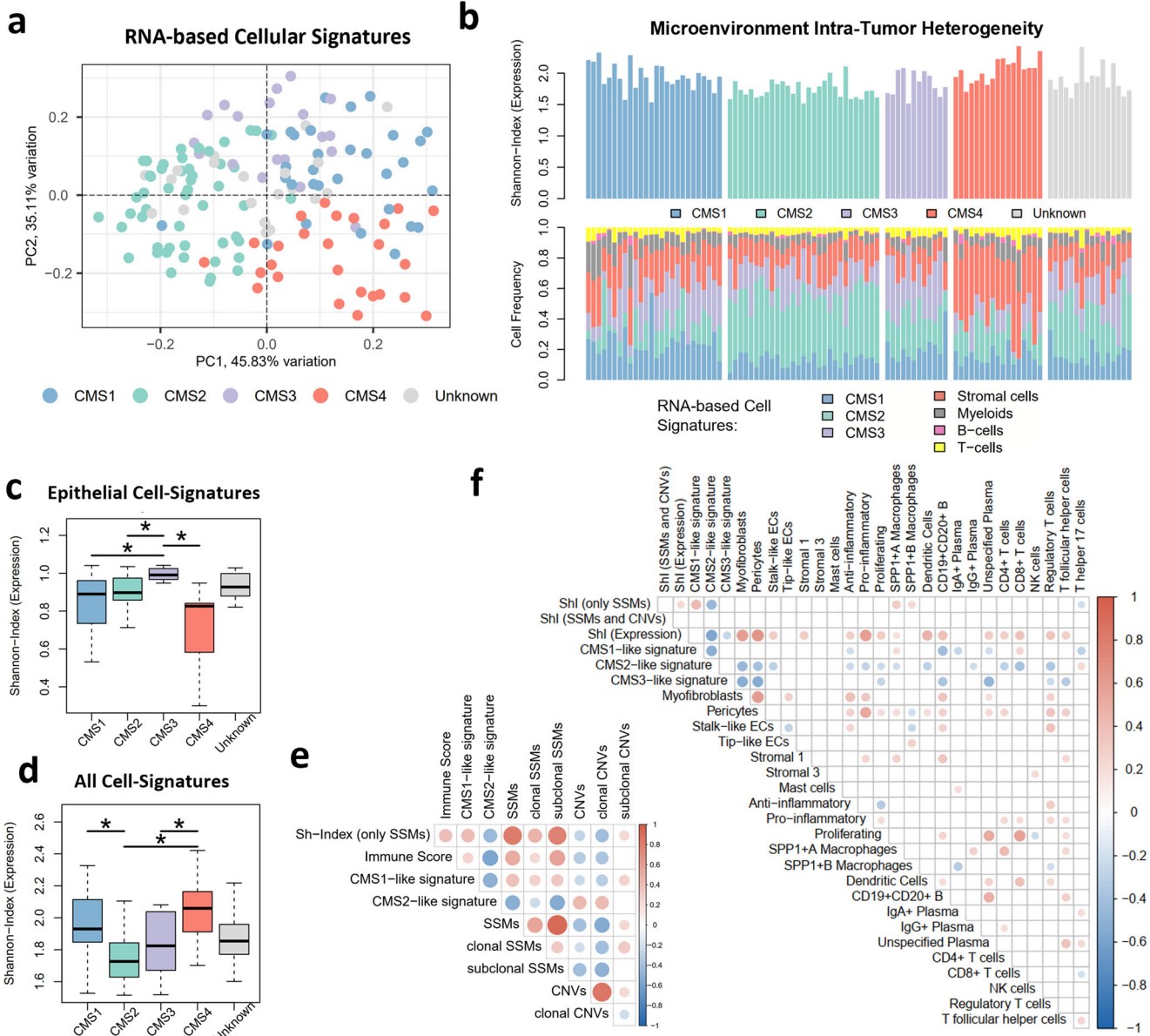

**Fig. 3 Microenvironment Inter- and Intra-Tumor Heterogeneity of CRC. a** Principal Component Analysis of cell frequencies from RNA-based deconvolution approach. CRC samples are colored according to CMS subtype. **b** Microenvironment composition of CRC samples grouped by CMS subtype, showing the microenvironmental Shannon-Index and cell frequency for RNA-based signatures of: epithelial cells (where CMS1, CMS2, CMS3-like are individually represented); stromal cells; myeloids; B-cells and T-cells. **c, d** Distribution of the microenvironment Shannon-Index (based on expression signatures) segregated by CMS subtype, considering the cell-frequency of: only epithelial cells (**c**) or all cell signatures (**d**). Numbers of samples per CMS subtype: CMS1 $n = 27$; CMS2 $n = 36$; CMS3 $n = 12$; CMS4 $n = 19$; Unknown $n = 18$. **e** Heatmap of Spearman correlation coefficient between: genetic Shannon-Index (only SSMs); immune score from Estimate; cell frequencies of CMS1/CMS2-like signatures; and number of SSMs and CNVs (total, clonal and subclonal). **f** Heatmap of Spearman correlation coefficient between genetic/microenvironmental Shannon-Indices and cell frequencies of the RNA-based signatures. Only significant correlations are represented (adj. $p$ value < 0.05).

a variety of CMS-like signatures seem to co-exist inside each tumor sample in our cohort (Fig. 3b and Supplementary Fig. 9a) and TCGA (Supplementary Fig. 9b). To quantify transcriptional ITH we applied the Shannon-Index, combining the number of distinct cell signatures found and respective frequencies (Fig. 3b). Interestingly, CMS3 tumors show the highest transcriptional heterogeneity when considering only epithelial cells in both cohorts, while CMS4 had lowest diversity (our cohort in Fig. 3c and TCGA in Supplementary Fig. 9c). Such results reinforce the epithelial signature of CMS3 subtype, and the established epithelial-to-mesenchymal transition common in CMS4 tumors[9]. However, the transcriptional Shannon-Index considering only

epithelial/stromal cells or also including the frequency of immune cells unveiled that CMS1 and CMS4 tumors contain a higher cellular diversity in both cohorts (our cohort in Fig. 3d and Supplementary Fig. 9d; TCGA in Supplementary Fig. 9e, f). In opposition, CMS2 tumors were the most homogenous, with a predominance of CMS2-like cell signature (our cohort in Fig. 3b and TCGA in Supplementary Fig. 9b). Concerning, MSI status or tumor location, increased microenvironment diversity was also observed for MSI-H and right-sided tumors (Supplementary Fig. 9g, h). Such results reinforce the segregation of CRC according to the initial genomic alterations, where tumors triggered by genomic instability develop more heterogeneous

microenvironments. Indeed, a low but significant correlation was found between cell diversity and genetic ITH assessed solely by SSMs in our cohort (Supplementary Fig. 10a, Spearman $r = 0.24$) and TCGA (Supplementary Fig. 10b, $r = 0.32$). Highly mutated tumors have been associated with more immune-cell infiltrates because these tumors carry more immunogenic mutations and harbor increased amounts of neoantigens[31]. In agreement, MSI-H tumors have higher immune scores (Supplementary Fig. 8b). Moreover, levels of genetic heterogeneity assessed solely by SSMs were significantly correlated with immune scores ($r = 0.39$, Fig. 3e).

Since distinct forms of genomic alterations foster genetic ITH in CMS1 and CMS2 subtypes, we next sought to evaluate if this feature is mirrored into the co-existent CMS-like signatures inside each primary tumor. Interestingly, the frequency of CMS1-like signature was correlated with genetic ITH (based on SSMs) when considering all primary tumors or even excluding the CMS1 subtype ($r = 0.41$, Fig. 3e and Supplementary Fig. 10c). In parallel, higher content of CMS2-like cell signature was associated with low mutational burden ($r = -0.51$) and high chromosomal instability for all primary tumors ($r = 0.42$), even in non-CMS2 tumors (Fig. 3e and Supplementary Fig. 10d). Indeed, mutations in the *APC* gene were associated with higher content of CMS2-like cell signatures in all primary tumors or non-CMS2 (Supplementary Fig. 10e, f), reinforcing the association between *APC* loss, chromosomal instability and CMS2 subtypes. Interestingly, CMS1 and CMS2-like signatures were negatively correlated and associated with distinct CRC microenvironments (Fig. 3f). High levels of CMS1-like signatures were slightly but significantly associated with higher frequencies of CD8 + T-cells ($r = 0.25$), whereas levels of Th17 cells seemed to correlate with CMS2-like signature ($r = 0.23$) (Fig. 3f). Several stromal ($r = 0.61$), myeloid ($r = 0.32$), and immune cells ($r = 0.34$) were associated with myofibroblasts, the marker for CMS4. In fact, the CMS1 subtype has been described as having a relatively stronger immune activation, particularly through the activity of adaptive immunity, namely Th1 and cytotoxic T-cells. CMS4 seems to be also associated with higher immune activation, but mostly through innate immunity, namely complement-mediated inflammation[9].

Further, we sought to model genetic heterogeneity through expression levels of cancer-related genes (Fig. 4a and Supplementary Fig. 11), where we could detect gene expression predictors of genetic diversity in our cohort and TCGA (Fig. 4b). The most frequently selected were *SMAD4* and *RBP3* genes, for which an increased expression is associated with higher heterogeneity.

Thus, our analysis suggests that genomic alterations underlying clonal diversity can determine microenvironmental inter- and intra-tumor heterogeneity.

**Genetic and microenvironmental intra-tumor heterogeneity influences metastatic potential.** To investigate the degree to which ITH can favor the evasion of tumor subclones and subsequent development of metastasis, we compared 20 early-stage (II–III) primary tumors that relapsed (mCRC) with 92 early-stage (II–III) primary tumors that did not develop distant (nmCRC) in 5 years of follow up after resection of the primary tumor. Consistent with previous research[9], metastasis development was mostly associated with stage III and the CMS4 subtype (Supplementary Fig. 12a–b). The CMS2 subtype also showed a nearly significant association with metastasis formations ($p$ value = 0.07), as the majority (23 out of 31) of metastatic primaries belong to CMS2 and CMS4 subtypes (Supplementary Fig. 12c). In our cohort CMS4 also showed a trend for lower survival probability (difference not significant relative to remaining subtypes,

Supplementary Fig. 12d), being consistent with the reported worse relapse-free and OS for CMS4 tumors[9].

Due to CRC high intra and inter-tumor heterogeneity, we next evaluated molecular features associated with metastatic potential, when gathering by MSI status, primary location and CMS classification (except for CMS1 and CMS3 given the low number of metastatic samples) (Fig. 5a). Surprisingly, metastasis development was associated with low genetic ITH (based on SSMs), when enclosing all primary tumors or just considering CMS2 and MSS tumors (Fig. 5a and Supplementary Fig. 13a–g). In fact, metastases were not developed in highly heterogeneous CMS2 tumors, contrasting with approximately half of the tumors with lower heterogeneity developing metastasis (Supplementary Fig. 14a). Equivalent differences were observed for MSS and left-sided tumors (Supplementary Fig. 14b, c). This suggests that the good prognostic value of increased ITH may be due to more immunogenicity of highly mutated tumors. However, such protective role can disappear in cases of excessive mutation burden and subclone diversity[11]. Indeed, segregating all primary tumors in four heterogeneity ranks, showed that extreme levels of genetic ITH (very low or very high) show higher probability to develop metastasis (Supplementary Fig. 14d). Nevertheless, CMS2 tumors with high or very high heterogeneity presented similarly a better relapse-free probability (Supplementary Fig. 14e), probably because CMS2 contains globally the lowest levels of clonal diversity. Therefore, our study shows that genetic ITH (based on SSMs) can discriminate CRC samples with high propensity to metastasize, where such prognostic role is more evident for CMS2 and MSS tumors.

Regarding chromosomal instability, our analysis showed that high CNVs burden in early primary tumors appears to favor metastasis development, expressly in MSS and left-sided tumors (Fig. 5a and Supplementary Fig. 13e, f). Indeed, relapsing tumors have an increased amount of CNVs (Supplementary Fig. 15a–c). When comparing the CNVs in early-stage primary samples that metastasize versus samples that do not, we observe a general increase in the frequency of CNVs (both amplifications and deletions) in samples that will give rise to metastasis, with a particular enrichment of amplifications in chromosome 6, encompassing cancer-associated genes such as *MYB* (Fig. 5b). There seems to be a stronger instability of chromosome 16 in primaries leading to metastasis, with enrichment of both amplifications and deletions. Other moderately enriched regions include the deletion of chromosome 18, an event widely reported to be associated with liver metastasis and poorer prognosis[32]. Besides *SMAD4* and *DCC*, this region encloses known cancer-related genes that were significantly associated with a higher relapse probability of early-stage primary tumors, namely *SMAD2* and *BCL2* (Supplementary Fig. 15d). Such genomic deletions lead to a depletion in *SMAD2* and *BCL2* transcripts levels (Supplementary Fig. 15e). *SMAD2* is an intracellular component of the TGF-β signaling pathway, which loss has been associated with CRC advanced-stage disease and shorter OS[33]. Furthermore, TGF-β activation in the TME promotes differentiation of myofibroblast which are associated with malignant progression of CRC[34]. *BCL2* encodes an anti-apoptotic protein implicated in CRC initiation, progression and therapy resistance[35].

Since CRC is differentially and asynchronously affected by genomic and chromosomal instability, we next sought to evaluate the prognostic value of SSMs and CNVs based on their clonality (Fig. 5c and Supplementary Fig. 16). Overall, metastatic potential was associated with low levels of clonal SSMs, especially for MSI-H and right-sided tumors, in line with the concept that highly mutated tumors are more immunogenic. Curiously, increased amounts of subclonal CNVs were associated with higher propensity to develop metastasis, particularly for MSI-H and right-sided tumors (Fig. 5c and Supplementary Fig. 16).

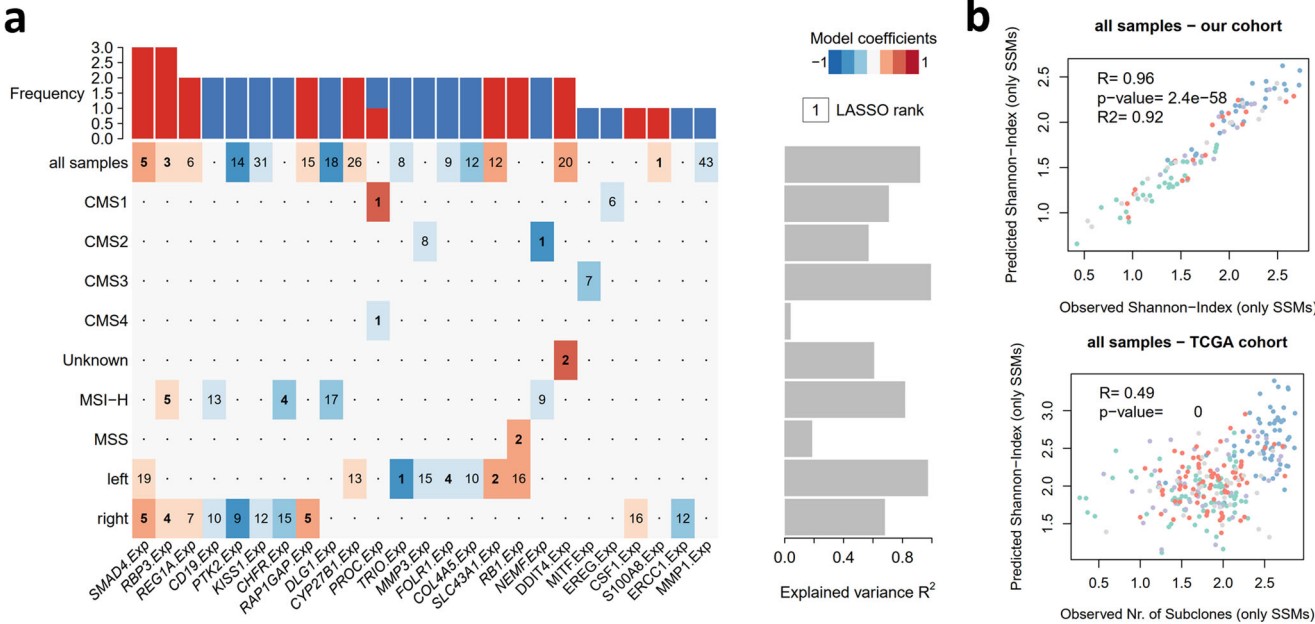

**Fig. 4 Transcriptional biomarker signatures for genetic ITH. a** Heatmap of cancer-related genes whose expression levels are associated with genetic ITH levels (only SSMs) in CRC samples depicted by a LASSO penalized model. Each line represents an independent analysis applied to the CRC samples segregated according to CMS subtypes (all samples $n = 112$; CMS1 $n = 27$; CMS2 $n = 36$; CMS3 $n = 12$; CMS4 $n = 19$), MSI status (MSS $n = 59$; MSI-H $n = 53$) or primary tumor location (right $n = 68$; left $n = 33$). LASSO-selected coefficients are colored according to the effect of each standardized covariate in the optimal model. The numbers on each tile denote the order in which variables are included indicating their relative importance. The top bar plot indicates the frequency at which each driver-gene mutation occurs in the ITH fitted model. The right bar plot shows the explained variance. **b** Comparison between observed and predicted genetic ITH levels (only SSMs) for all samples in our cohort and in TCGA.

Besides genetic alterations, low microenvironmental heterogeneity (Shannon-Index from RNA-based cellular deconvolution) appears also to favor metastasis development (Fig. 5a and Supplementary Fig. 13a–g), suggesting a crucial role for TME richness. A comprehensive analysis of the microenvironment heterogeneity using transcriptional cell signatures unraveled that metastatic potential was associated with low content of stalk-like endothelial cells and dendritic cells, and high amount of myofibroblasts and pro-inflammatory macrophages (Supplementary Fig. 17a). Notably, segregating the early primary tumors based on the ratio of such pro and anti-metastatic cell frequencies, we could observe a significant difference in relapse-free and OS (Supplementary Fig. 17b).

Therefore, our analysis indicates that genetic and microenvironment ITH can determine the propensity of a primary tumor to relapse, emphasizing the determinant role of mutational and chromosomal instability in tumor progression.

**Clonal diversity driven by CNVs favors metastasis development**. To evaluate how clonal diversity would evolve during metastatic formation, we compared 112 CRC primary tumors and 12 metastases. Overall, we found lower genetic ITH in metastasis (Fig. 6a), supporting that metastasis development imposes a bottleneck in clonal diversity[36]. Since chromosomal instability fosters metastatic potential, we reasoned that the CNVs-rich subclones escaping from the primary site would originate metastasis with a higher content of clonal CNVs. Indeed, metastasis showed a higher content of CNVs burden and aneuploidy, and more specifically of clonal CNVs (Fig. 6b–d and Supplementary Fig. 18a). Conversely, SSMs burden was higher in primary tumors for clonal or subclonal alterations (Supplementary Fig. 18b). Such results are consistent with a previous pan-cancer study showing that CNVs are more frequently clonal in metastases compared with primary tumors[37]. Regarding the TME

composition, metastasis showed higher content of anti-inflammatory macrophages and CD4+ T-cells (Supplementary Fig. 18c). Consistently, higher amount of anti-inflammatory marker cells (CLEVER-1/Stabilin-11+ cells) in metastatic setting correlated with shorter disease-free survival[38].

Next, we sought to deeply explore tumor evolution by inspecting 12 pairs of primary tumor and distant liver/lung metastasis. First, we inferred the clonality of individual metastases through the Jaccard Similarity Index (JSI) that considers the number of shared and private SSMs, with higher values corresponding to multiple dissemination events (polyclonal if JSI > 0.4)[39,40]. Most metastases were monoclonal and only three cases resemble scenarios of metastasis formation by multiple subclones (Supplementary Fig. 19a). Due to the higher prevalence of CNVs in metastasis development we also re-estimated the JSI enclosing both SSMs and CNVs, which did not seem to strongly affect the clonality status (Supplementary Fig. 19b). A more comprehensive analysis, inferring tumor clonal phylogenies of each primary-metastasis pair[20], unraveled distinct evolutive branches of primary and metastasis progression, whereas the JSI-polyclonal metastasis showed more than one founder subclone (Fig. 6e, f and Supplementary Fig. 19c, d). Analysis of additional 19 CRC primary-metastasis pairs publicly available[41] reinforced monoclonal and polyclonal seeding in CRC progression (Supplementary Fig. 20). As expected, polyclonal metastases were associated with high genetic clonal diversity in primaries and metastasis from both paired cohorts (Supplementary Fig. 21a). Moreover, such multiple seeding events appear to occur later in tumor progression, given the positive correlation found with subclonal genetic alterations (Supplementary Fig. 21a). Interestingly, polyclonal seeding was associated with the CNVs-rich CMS2-like signature in metastasis, reinforcing the contribution of chromosomal instability for metastasis development (Supplementary Fig. 21a).

Finally, to assess for possible events leading to metastasis, we evaluated CNVs shared between paired primaries and metastasis

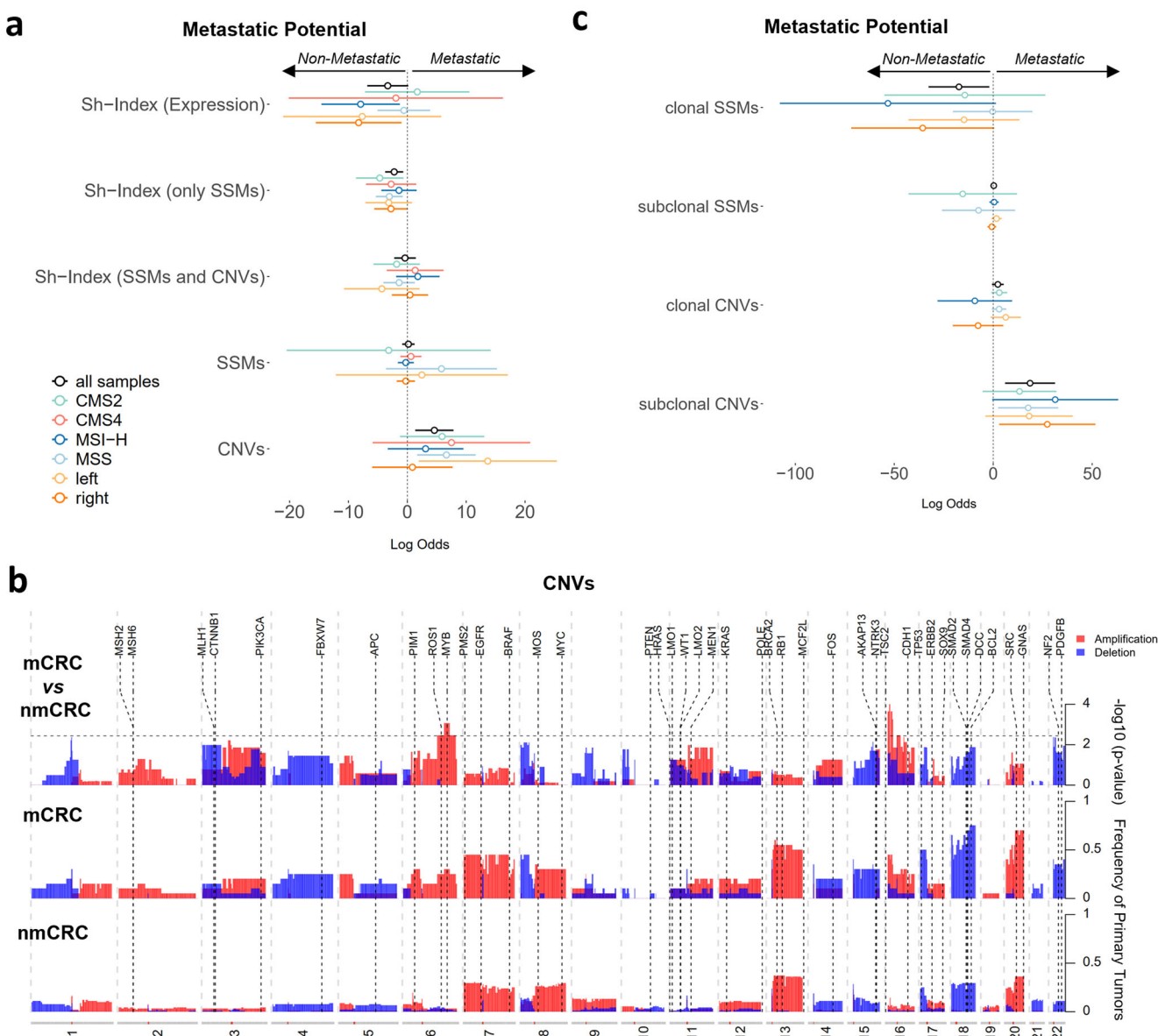

**Fig. 5 Tumor heterogeneity and Metastatic potential. a** Coefficients (log-odds ratios) of generalized linear models for metastatic potential including intra-tumor heterogeneity (genetic and microenvironment); number of genomic alterations (SSMs and CNVs). Each color represents an independent model fitted to the CRC samples segregated according to CMS subtypes (all samples $n = 112$; CMS1 $n = 27$; CMS2 $n = 36$; CMS3 $n = 12$; CMS4 $n = 19$), MSI status (MSS $n = 59$; MSI-H $n = 53$) or primary tumor location (right $n = 68$; left $n = 33$). Detailed results and significance levels are represented in Supplementary Fig. 13. **b** Frequencies of copy number events (separated in amplifications and deletions) affecting different regions of the genome (binned by chromosomal bands), in early primaries that do not metastasize (nmCRC, $n = 92$), and in early primaries that metastasize (mCRC, $n = 20$). The top panel shows the enrichment of event frequency in the metastatic versus non-metastatic primaries (values represent the $-\log 10$ of the uncorrected $p$ value of a fisher test for each genomic bin). Positions of known cancer-related genes are also displayed. **c** Coefficients (log-odds ratios) of generalized linear models for metastatic potential including clonal and subclonal genomic alterations (SSMs and CNVs). Each color represents an independent model fitted to the CRC samples segregated according to CMS subtypes, MSI status or primary tumor location. Detailed results and significance levels are represented in Supplementary Figs. 15 and 16.

in both paired cohorts (Fig. 6g and Supplementary Fig. 21b). Among the most common shared CNVs, we find gains of chromosomes: 7 (encompassing *PMS2, EGFR* and *BRAF* genes), 8q (*MYC*), 13q (encompassing *BRCA2* and *RB1* genes) and 20q (enclosing *SRC* and *GNAS*). Moreover, we also found a recurrent loss of chromosomes: 8p, 17p (enclosing *TP53*) and 18 (encompassing *SMAD2, SMAD4, DCC* and *BCL2*).

Overall, our work unveils that chromosomal instability plays an important role in the metastasis process, boosting the creation of new tumor subclones and favoring multiple seeding.

## Discussion

In this study, we combined genomic and transcriptomic profiles of CRC samples to evaluate how the different genomic instability forms can shape inter and ITH, and consequently determine CRC progression and metastasis development.

The analysis was performed in a prospective collection of samples in stage II and III CRC patients with curated clinical information regarding relapse-free survival (RFS) and OS. Nonetheless, we acknowledge limitations due to small sample size, particularly given the small number of metastasizing tumors

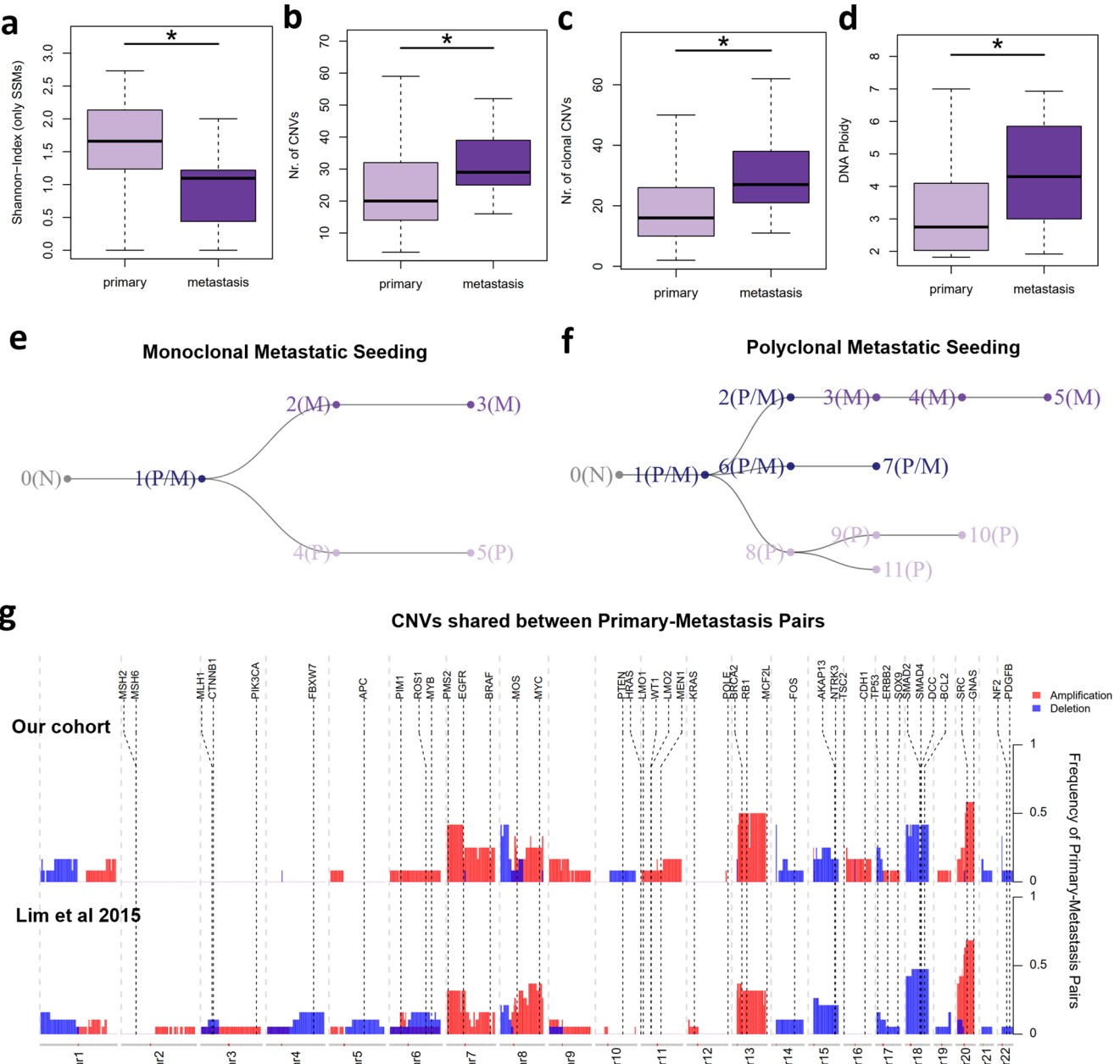

**Fig. 6 Clonal Diversity and Metastasis Development. a** Distribution of the Shannon-Index values based on Expands clonal composition for early primary tumors ($n = 112$) and metastasis ($n = 12$). **b–d** Amount of total CNVs (**b**), clonal CNVs (**c**) and DNA ploidy (**d**) for early primary tumors and metastasis. **e**, **f** Phylogenetic trees depicting subclonal expansion for a primary-metastasis pair with monoclonal (**e**) and polyclonal metastatic seeding (**f**). The subclones are identified with the respective number and the containing sample: normal (N), primary tumor (P) and metastasis (M). **g** Frequencies of copy number events (separated in amplifications and deletions) affecting different regions of the genome, in the paired primary-metastasis samples from our study ($n = 12$) and the study of Lim et al. ($n = 19$). Only events found in both the primary and its paired metastasis are displayed. Positions of known cancer-related genes are also displayed.

which reduce statistical power. Nevertheless, we reinforced our findings using independent cohorts of early primary tumors from TCGA[3] and primary-metastasis pairs publicly available[41]. We also acknowledge that assessing ITH from bulk genomic profiles is intrinsically difficult, where the different available tools can provide inconsistent measures[20]. Moreover, bulk sequencing of a single region is likely to provide underestimated measures of ITH when compared with multi-region or single-cell studies. Nonetheless, we used two different tools, Expands and PhyloWGS, from which different ITH measures were obtained. Given that Expands relies mostly on SSMs, it is not surprising that the

hypermutable CMS1 subtype showed the highest ITH levels, while CMS2 was the most homogeneous. Conversely, since evolution of CMS2 tumors is mainly driven by chromosomal instability, their actual clonal diversity was only disclosed when including both SSMs and CNVs with PhyloWGS. Such results highlight the importance of selecting appropriate methods to avoid misleading estimates of ITH for tumor samples with high chromosomal instability and low mutation rate.

Despite CMS stratification being associated with distinct forms of genomic alterations, our analyses depicted pervasive ITH in all CMS subtypes. More importantly, clonal diversity in CRC results

from a jointly and asynchronous action of different forms of genetic lesions. CMS1 ITH appears to be fostered by MSI, but it is subsequently sustained by accumulation of both SSMs and CNVs. Such findings are consistent with the common evolutionary scenario where MSI is induced in the early-stages of the primary tumor, followed by increased genomic burden and focal copy-number alterations[21]. However, some CRC tumors, namely CMS2 tumors, exhibit a different evolutionary path, where chromosomal instability events emerge early. Furthermore, the inverse correlation observed between SSMs and CNVs levels emphasizes the existence of a threshold for tolerable genomic instability, above which the excessive accumulated genetic lesions compromise cellular viability[20,21].

Our analysis revealed that the distinct molecular pathways forging tumor evolution will also determine the TME composition. In fact, CRC tumors triggered by mutational instability developed more heterogeneous microenvironments, enriched in immune cell content. This is in agreement with highly mutated tumors harboring more immunogenic mutations and consequently attracting more immune-cell infiltrates[31]. In fact, a recent study suggests that the CMS1 and CMS4 signal is mostly derived from immune and stromal cells, respectively[42], with little contribution from tumor epithelial cells. Besides the differential content in stromal and immune cells, CRC tumors also showed ITH at the level of epithelial cells. Indeed, a variety of CMS-like signatures were detected inside each tumor sample, as previously described[30,42]. As expected, the mesenchymal subtype CMS4 showed the lowest epithelial ITH, while CMS3 tumors harbored the highest levels of each CMS-like cell. More importantly, the individual CMS1- and CMS2-like cell frequencies inside each tumor were associated with increased mutational and chromosomal instability, respectively. Such findings reinforce the link between specific genomic alterations and CRC transcriptomic phenotypes, highlighting the diversity of tumor subclones within each tumor sample. Furthermore, we identified predictive signatures using genetic and expression profiles of cancer-related genes that can foresee ITH levels in an independent CRC cohort from TCGA, regardless of the tumor subtype, MSI status or sidedness. Such gene signatures are prominent ITH biomarkers that can be used to monitor clonal diversity. Moreover, the genes composing such signatures are potential drivers of subclonal expansion, which causal effect can be experimentally confirmed, for further use as therapeutic targets. The poor capacity of our gene signatures to predict PhyloWGS-based ITH levels in TCGA data could be due to the different platforms used to detect CNVs in both cohorts, which would affect the estimation of tumor subclones by PhyloWGS.

Beyond CMS stratification, molecular and histological differences have also been observed between right and left-sided CRCs[18,43]. Our study extends these findings to demonstrate that primary tumor location is associated with distinct genetic and microenvironment heterogeneities; with right- and left-sided tumors resembling CRC tumors triggered by microsatellite and chromosomal instability, respectively. Given the fact that two sides of the colon have distinct embryologic origins and cell differentiation processes, this suggests that such basal development programs probably contribute to CRC diversity[44]. A recent study using multi-region sequencing of 68 patients confirmed this distinction, suggesting a higher complexity of left-sided tumors[45].

Previous work has shown that levels of genetic ITH could predict metastatic CRC, when considering also advanced tumor stages[16]. Here, we evaluated how genetic and microenvironmental heterogeneity could influence metastatic potential in early-stage primary tumors (Stage II/III), enclosing the distinct CRC subgroups. Interestingly, metastasis development was associated with low genetic ITH (based on SSMs), more explicitly in MSS CMS2

tumors. Such findings are consistent with the lower mutation burden found in relapsing CRC[46]. A favorable outcome can be explained by the high levels of immune cells found in highly heterogeneous tumors. Accordingly, immune-cells infiltration in primary tumors has been associated with reduced CRC tumor dissemination[42,47]. However, the protective capacity of the immune system seems to be overrun by cases of excessive clonal diversity. Such findings are consistent with a previous pan-cancer analysis that also observed a nonlinear association between ITH and survival[11]. In parallel, our analysis of microenvironmental heterogeneity also supports the important role of a prosperous TME to counter metastatic potential. Notably, we were able to identify specific pro and anti-metastatic RNA-based cell signatures, which were associated with worse relapse-free and OS. Although, tumor-associated endothelial cells have usually been associated with the promotion of tumor growth and resistance to treatment[48], our results suggest that at least a subset of tumor-associated endothelial cells may have a protective role in preventing relapse. Similarly, a dual role has been also described for dendritic cells by promoting tumor invasiveness and worse outcome[49] or association with immune activation and clearance[50]. Indeed, our analyses associate general markers of dendritic cells with less prevalence of relapse. On the other hand, enrichment of myofibroblasts and pro-inflammatory macrophages were shown to induce pro-tumorigenic and immunosuppressive mechanisms that lead to tumor progression and metastases[51]. In fact, the presence of myofibroblasts has been previously associated with tumor invasiveness in colorectal cancer[52–54]. Other cells from the innate and adaptive immune system have been associated with metastatic potential[55], however due to their low frequency in our RNA-based deconvolution data it was not possible to accurately explore their association. Besides genetic and TME features, expression levels of human endogenous retrovirus have also been associated with more aggressive subtype of CRC[56]. Nevertheless, our microenvironment signatures are potential biomarkers to predict metastatic potential and guide treatment choices based on biopsies from early-stage primary tumors.

Since metastasis is the main cause of CRC deaths, and CMS, MSI and tumor location although prognostic markers, are not enough to effectively assess patients´ outcome, genetic ITH adds predictive value of metastasis development in CRC patients. In our study we unveil that late-emerging CNVs in early stage CRCs can potentiate metastasis formation and multiple events of metastatic seeding. In fact, chromosomal instability and high CNVs levels have been associated with worst clinical outcome in several cancers[57–59], because they can rapidly generate a diversity of cellular phenotypes required for the complex process of metastatic spread. In CRC, CNV heterogeneity has been associated with acquisition of advantageous traits that would favor tumor progression and metastasis[19,57]. A recent study with 84 untreated stage II CRC patients showed that a higher burden of CNAs, particularly subclonal, was associated with an increased proportion of recurrence[60], while the same was not found for SNVs. Our results are also consistent with a recent pan-cancer study, showing that late-emerging subclones may seed metastasis[37]. Also, a single-cell genomic profile of primary-metastasis pairs of two CRC patients, revealed a late dissemination model with polyclonal seeding[61], with results from another study also suggesting that metastasizing tissues are often formed from multiple clones[45]. Overall, such findings support an important role of chromosomal instability in metastasis development, where CNVs burden can be used as marker of risk of metastasis[57,60].

Collectively, our analyses provide a detailed picture of the molecular alterations shaping genetic and microenvironment heterogeneity between and within CRCs. By integrating genomic

and transcriptomic profiles, we were able to show how the interplay between genomic instability pathways and cellular phenotypes can influence tumor evolution and metastasis development. Recent advances in single-cell profiling and spatial technologies shed more light on the complexity of the tumor ecosystem[62,63]. Indeed, orthogonal integration of single-cell and spatial transcriptomics from human squamous cell carcinoma samples allowed the dissection of cellular organization and communication within TME[63]. These emerging technologies have the potential to elucidate CRC inter- and intra-tumor diversity, with significant implications in the choice of specific molecular biomarkers and clinical decision-making.

## Methods

**Patient cohorts.** CRC samples encompassing primary tumors and adjacent normal colonic mucosa ($n = 124$) or metastasis ($n = 12$), were collected by Medical Pathologists from surgically removed colon specimens. The cohort enclosed: 20 early stage II–III primary tumors with metastatic relapse (mCRC); 92 early stage II–III primary tumors that did not develop distant metastasis (nmCRC) and 12 pairs of primary-metastasis samples. Normal mucosa samples were taken more than 2 cm away from the tumor. Tissues were embedded in optimal cutting temperature medium, snapshot frozen in liquid nitrogen within 40 min of collection and preserved at −80 °C. Samples were collected between June 2010 and October 2017 as part of a prospective biobanking project. For the entire cohort, median age was 70 years (min–max 36−91); 68 (61%) patients were female; 65 (58%) patients were diagnosed with stage II and 47 (42%) with stage III CRC, according to AJCC[64]. Patients had not received neoadjuvant chemo or radiotherapy (clinico-pathologic features are summarized in Supplementary Table 1). We defined RFS and OS as follows: for RFS, survival time between cutoff date for follow-up or censoring date, or study end point date (i.e., death/relapse date or last follow-up) and date of surgery; for OS survival time between survival time between cutoff date for follow-up or censoring date, or study end point date (i.e., death/relapse date or last follow-up) and date of diagnosis.

The study was approved by the ethics committee from Hospital de Santa Maria, Centro Hospitalar Universitário Lisboa Norte (Lisbon, Portugal) and all patients provided signed informed consent.

Furthermore, clinical data and multi-omics profiles from 126 early-stage CRC patients of The Cancer Genome Atlas (TCGA) project[3] and 20 paired primary and liver metastases CRC samples were included[41].

**Nucleic acid extraction and quality control.** DNA and RNA were extracted from three consecutive 30 μm cryosections, using the AllPrep® DNA/RNA Micro Kit (Qiagen), following the manufacturer's protocol. Presence of CRC cells in tumor samples and absence in normal tissues were confirmed in H&E sections by a Medical Pathologist. Extracted DNA was quantified with Qubit™ dsDNA High Sensitivity (Thermo Fisher Scientific). DNA quality was determined by the delta-Cq method (Illumina® TruSeq™ FFPE DNA Library Prep QC Kit); samples were required to have delta-Cq < 6 for library preparation. Extracted RNA was quantified with Qubit™ RNA High Sensitivity (Thermo Fisher Scientific). RNA quality was determined by the DV200 method using Agilent RNA 6000 Pico Kit (Agilent Technologies); samples were required to have a DV200 > 40% for library preparation.

**Whole transcriptome sequencing.** Illumina TruSeq™ Stranded Total RNA (Illumina, PN 20020597) with 100 ng input per RNA sample was used for generating whole transcriptome libraries according to manufacturer recommendations. Integrated DNA Technologies (IDT) for Illumina TruSeq™ RNA UD Indexes (96 indexes, Illumina PN 20022371) were used for sample indexing. Libraries were quantified with Qubit™ dsDNA High Sensitivity assay and normalized for sequencing on Illumina NovaSeq™ 6000 S2 (36-plex) or S4 (72-plex) flow cell with 76 bp paired-end sequencing targeting ~200 million read pairs per sample.

**Whole exome sequencing.** Illumina Nextera™ Flex for Enrichment (Illumina, PN 20025520) with 40 ng input per sample was used for generating matched tumor and normal enriched libraries, with some modifications. IDT for Illumina Nextera™ DNA Unique Dual Index Set A (PN 20027213) was used for sample indexing with 9 cycles of indexing PCR. Samples were quantified with Qubit™ dsDNA High Sensitivity assay and 4 libraries were pooled for enrichment (4-plex) such that 500 ng of each library was used for a total of 2000 ng per enrichment pool. Target enrichment was performed using IDT™ xGen Exome Research Panel v1 (4 μl per enrichment reaction). A single hybridization was done overnight at 58 °C, with 12 cycles of post-enrichment PCR. Libraries were quantified by Qubit™ dsDNA High Sensitivity assay, normalized and pooled for sequencing 12-plex per lane on NovaSeq™ 6000 S4 flow cell using the XP workflow for individual lane loading. Sequencing reads were 151 bp paired-end, providing a median coverage per locus of 118x (after filtering).

**Genomic profile analysis.** Somatic variants were obtained using a method akin to the broad best practices[65]. In brief, whole exome sequencing reads were aligned using bwa mem[66] against the human hg38 genome from the GATK resource bundle. Duplicates were removed using samblaster[67], and base qualities were recalibrated using GATK. Simple Somatic Mutations (SSMs) were obtained using the Mutect2 method[68]. To reduce noise, we only considered SSMs with VAF > 5% and with potential impact (annotated using annovar[69]) in protein coding genes. Clonality of SSMs was determined from their estimated cancer cell fractions as previously described[39]. Tumor-specific copy number estimates were obtained using cnvkit[70] by comparing alignments of each tumor sample against a pool of normal samples. Clonality of CNVs was obtained from absolute[71] estimates. For the TCGA cohort, SSMs (from MuTect) and microarray-based CNVs were downloaded from the Genomic Data Commons portal.

**Transcriptome profile analysis and RNA-based deconvolution.** Transcript abundances were estimated from RNA-sequencing reads using kallisto[72] against gencode (v32) transcripts and normalized to Counts per Millions (CPMs) with the edgeR package[73]. Gene set enrichment analysis of CMS signatures was obtained using the fgsea package[74] with signatures from the CMS caller R package[75], where genes were ranked according to the moderated T-statistics from the limma package[76], based on voom normalized expression values[77]. RNA-based cellular deconvolution of the expression values of CRC samples was performed with CIBERSORTx[78], using as cellular signatures the RNA-Seq TPM values and CRC cell-type annotations from the single-cell study of Lee et al.[30]. Normalized expression values (CPMs) of CRC samples were used to obtain stromal and immune scores from Estimate[79]. For the TCGA cohort, expression levels were obtained from the HTseq counts downloaded from the Genomic Data Commons portal and processed similarly to the samples from our cohort.

**Molecular classification of CRC samples.** We used three different algorithms to determine the Consensus Molecular Subtypes from RNA-Seq profiles: CMS caller[75]; CMS Classifier[9] and CMS String[80]. The final CMS type was attributed based on the consistent classification by at least two tools, or "Unknown" otherwise. MSI status was determined from whole-exome sequencing data using MSISensor2[81]. MSI status for the TCGA cohort was obtained from a previous work[20].

**Estimation of intra-tumor heterogeneity.** Genetic ITH was estimated using two independent approaches: solely based on SSMs using Expands[11]; or combining both SSMs and CNVs using PhyloWGS[20]. The scripts used to estimate genetic ITH can be found at GitHub: https://github.com/comicsfct/OncoHIT/. Each tool provides the number of subclones and respective cellular prevalence inside each tumor sample, which were used as input metrics to estimate cellular diversity applying the Shannon-Index[17]. The Shannon-Index is a simple and widely used measure of diversity, with several other measures being simple variations of the same formula. There is a vast literature on measuring diversity, and other approaches might give a better representation of rare populations, such as Hill diversity[82], but given that these rare clusters are more likely to derive from noise, the Shannon-index should provide a conservative balance. Only samples with tumor purity > 0.4 were considered for comparison of genetic heterogeneity levels. For the TCGA cohort, Expands was applied using SSMs (from Mutect) and microarray-based CNVs downloaded from the Genomic Data Commons portal. Clonal composition based on SSMs and CNVs of TCGA samples was obtained from the original PhyloWGS study[20]. The JSI was calculated as described in[39] using the variants and clonality obtained. Microenvironmental ITH was also estimated through the Shannon-Index, combining the number of distinct cell signatures and respective frequencies found with the RNA-based deconvolution approach.

**Biomarkers for intra-tumor heterogeneity.** To identify biomarkers of genetic ITH we used Lasso penalized regression modeling as previously applied[22,83]. Briefly, the Shannon-Index obtained from the clonal frequencies estimated for each sample was used as a dependent variable, and different genetic events were used as independent variables. We then applied Lasso regression, as implemented in the glmnet R package[84], with a tenfold cross validation to choose the lambda parameter. In order to reduce the noise created by passenger or silent genetic alterations, we only considered SSMs with predicted functional impact or CNVs affecting 2174 cancer-related genes from Cancer Genetics Web (https://www.cancer-genetics.org/#download) and CancerQuest (https://www.cancerquest.org/cancer-biology/cancer-genes#1). Such focused analysis in cancer-related genes was applied to reduce the number of features (avoid overfitting) and exclude genes that may not be biological relevant for cancer. Furthermore, to eliminate an over-representation of hypermutated genes in the model, SSMs with VAF higher than 5% were converted into a binary matrix representing the presence/absence of mutations for each gene on each tumor sample. CNVs were resumed to values representing deletions (−1), amplifications (+1) and none (0). For transcriptomic biomarkers we used the normalized expression levels (logCPMs) of cancer-related genes. Finally, the fitted models were evaluated by comparing the observed and predicted ITH levels based on the tumor mutation profiles of our cohort and TCGA, through Pearson correlation coefficient.

**Statistics and reproducibility**. Figures were produced using ggplot2 R package and default packages from the R environment. Heatmaps were produced with pheatmap and corrplot R packages. Principal component analysis was performed and graphically represented with the PCATools R package. Regression modelling was applied using glm function from stats R package and displayed using forestplot and jtools packages. The oncoprint plot was produced using the ComplexHeatmap R package. Survival and relapse-free curves were analyzed by Kaplan–Meier curve comparison using a log-rank test and with a multivariate Cox proportional hazards analysis as implemented in the survival and survminer R packages. The tumor evolutive trees were reproduced using networkD3 R package. The statistical significance of differences between groups was evaluated using Wilcoxon test ($*p < 0.05$; $**p < 0.01$; $***p < 0.001$; $****p < 0.0001$). The sample sizes for each group were: CMS subtypes (all samples $n = 112$; CMS1 $n = 27$; CMS2 $n = 36$; CMS3 $n = 12$; CMS4 $n = 19$), MSI status (MSS $n = 59$; MSI-H $n = 53$); primary tumor location (right $n = 68$; left $n = 33$); early primaries that do not metastasize (nmCRC, $n = 92$), and early primaries that metastasize (mCRC, $n = 20$).

**Reporting summary**. Further information on research design is available in the Nature Research Reporting Summary linked to this article.

## Data availability

The datasets generated and/or analyzed during the current study are under accession numbers PRJNA689313 (National Center for Biotechnology Information) and EGAD00001007686 (European Genome-Phenome Archive). The source data used to generate the main figures is provided as Supplementary Data.

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

## Acknowledgements

We thank João Neto and remaining Computational Multi-Omics Lab members at UCIBIO for their helpful comments and suggestions. This work was financed by national funds from FCT - Fundação para a Ciência e a Tecnologia, I.P., in the scope of the project UIDP/04378/2020 and UIDB/04378/2020 of the Research Unit on Applied Molecular Biosciences - UCIBIO and the project LA/P/0140/2020 of the Associate Laboratory Institute for Health and Bioeconomy - i4HB. This research was also funded by: PTDC/MED-ONC/28660/2017 from Fundação para a Ciência e Tecnologia (FCT) to A.R.G. A.R.G is recipient of Researcher Grant CEECIND/02699/2017 from FCT. The bio-banking of CRC samples from Hospital Santa Maria, Lisbon, Portugal was supported by FCT research grant PIC/IC/82821/2007. This work was produced with the support of INCD funded by FCT and FEDER under the project 22153-01/SAICT/2016.

## Author contributions

M.M. was responsible for the DNA and RNA extraction of CRC samples. S.K. performed the genomic and transcriptomic sequencing profiles. D.S., R.P., and A.R.G. conducted the bioinformatics analyses. P.M.C., R.E., F.A., A.R., J.C., A.Q., and C.F. were responsible for the surgical collection of CRC samples. M.M. and S.C. were responsible for the management and storage of samples under the CRC biobank. A.F. and P.B. were responsible for the histopathological characterization of CRC samples. C.A., A.L.C., D.M., S.M., P.F., and A.M. were responsible for clinical data collection. M.S., N.K., R.V., S.Z., T.P., and J.G. performed WES and WTS. M.G., A.S., and L.L. are responsible for the critical revision of the paper and for important intellectual content. D.S., M.M., A.R.G., and L.C. conceived and designed the study, interpreted the data and wrote the paper. All authors read and approved the final paper.

## Competing interests
The authors S.K., M.G., M.S., N.K., R.V., S.T., T.P., J.G., A.S., L.L. are current employees and shareholders of Illumina Inc. The authors declare that they have no competing interests.

## Ethics approval and consent to participate
The study was approved by the ethics committee of Hospital de Santa Maria, Centro Hospitalar Universitário Lisboa Norte (Lisbon, Portugal) and all patients provided signed informed consent.
