## [Peer Review File · Communications Biology]

Reviewers' comments:

Reviewer #1 (Remarks to the Author):

Sobral and coworkers have explored the intra-tumoral heterogeneity of colorectal cancer, showing some molecular mechanisms involved in tumor evolution and metastasis. This study was based on whole exome sequencing and RNA sequencing data of 124 primary tumors and 12 matched, metastatic tumors and on the analysis of data deriving from the TCGA databases. The genetic heterogeneity was analyzed through analysis of the clonality of somatic mutations and of the copy number variations; furthermore, it was explored the heterogeneity of tumor microenvironment by CIBERSORT. To quantify the degree of intratumor heterogeneity they have used the Shannon-Index. The authors confirm many data reported in the literature and have made four main observations: (i) the burden of copy number variations is associated with metastases; (ii) somatic mutations and copy number variations asynchronously contribute to clonal diversity; (iii) the metastatic potential is associated with the level of genetic and immune intra-tumor heterogeneity; (iv) different CSM classes are associated with a variable immune profile.

The authors have provided a satisfactory reply to criticisms/comments/suggestions received in a previous round of reviewing.

- The authors should better discuss in the first paragraph of the discussion that their study is based on a methodology that may provide an underestimation of intra-tumor heterogeneity, compared to studies based on multiregion WES or single-cell sequencing (genomics and transcriptomics).
- The authors should mention and to discuss the following papers: (i) Lahoz S et al. *Journal of Pathology* 2022; 257: 68-81.; (ii) Chowdhury S et al, *Cancers* 2021; 13: 4923.; (iii) Banerjee S et al. *iScience* 2021; 24: 102718.

Reviewer #2 (Remarks to the Author):

Sobral et al. have presented a comprehensive analysis of the genetic and microenvironmental heterogeneity of colorectal cancer using a prospective biobank of primary and metastatic tumor tissues (including matched pairs), compared to publicly available data sets. Specifically, they highlight unique features of intra-tumoral heterogeneity that favor tumor progression and metastasis. The authors have revised the manuscript to outline the advantages of their findings more clearly over prior publications. They have made additional changes in response to prior reviewer comments that strengthen the overall work. There are a few minor edits that would improve the manuscript.

1. Line 86 – remove “ed” on developed
2. Lines 256-258 – is there an explanation for why KRAS mutations were not detected in CMS3 samples? Has this been observed in other studies?
3. Line 387 – change to “CMS classification” rather than CRC classification
4. Line 658 – change to “A favorable outcome” rather than The favorable
5. Line 677 – can the authors provide references that support the microenvironmental signatures (e.g., endothelial stalk cells, dendritic cells, fibroblasts) postulated as biomarkers for metastatic potential? There are several colorectal cancer spatial transcriptomic and single cell RNAseq experimental data sets published that could be useful to look at and cite (e.g., <https://doi.org/10.1016/j.ccell.2022.02.013>; <https://doi.org/10.1038/s41467-022-29366-6>).
6. Line 696 – I would suggest making this a separate section “Limitations of Study” to acknowledge

the small study sample size and include the limitations for assessing ITH using bulk sequencing analysis vs single cell analyses.

RESPONSE LETTER

Genetic and microenvironmental intra-tumor heterogeneity impacts colorectal cancer evolution and metastatic development

We thank the Referees for their thorough review of our manuscript and their constructive comments and suggestions. We present herein our point-by-point rebuttal for your evaluation and the revised manuscript (alterations highlighted in red).

Point-by Point response to Reviewer #1

Sobral and coworkers have explored the intra-tumoral heterogeneity of colorectal cancer, showing some molecular mechanisms involved in tumor evolution and metastasis. This study was based on whole exome sequencing and RNA sequencing data of 124 primary tumors and 12 matched, metastatic tumors and on the analysis of data deriving from the TCGA databases. The genetic heterogeneity was analyzed through analysis of the clonality of somatic mutations and of the copy number variations; furthermore, it was explored the heterogeneity of tumor microenvironment by CIBERSORT. To quantify the degree of intratumor heterogeneity they have used the Shannon-Index. The authors confirm many data reported in the literature and have made four main observations: (i) the burden of copy number variations is associated with metastases; (ii) somatic mutations and copy number variations asynchronously contribute to clonal diversity; (iii) the metastatic potential is associated with the level of genetic and immune intra-tumor heterogeneity; (iv) different CSM classes are associated with a variable immune profile.

The authors have provided a satisfactory reply to criticisms/comments/suggestions received in a previous round of reviewing.

Response: We would like to thank the reviewer for her/his careful evaluation of our manuscript and appreciate the overall positive comments.

5. The authors should better discuss in the first paragraph of the discussion that their study is based on a methodology that may provide an underestimation of intra-tumor heterogeneity, compared to studies based on multiregion WES or single-cell sequencing (genomics and transcriptomics).

Response: We understand the reviewer's comments and have rewritten the first paragraph of the Discussion to discuss the limitations of our study (lines 436-442).

Rewritten text:

“Nonetheless, we acknowledge limitations due to small sample size, particularly given the small number of metastasizing tumors which reduce statistical power. Nevertheless, we reinforced our findings using independent cohorts of early primary tumors from TCGA [3] and primary-metastasis pairs publicly available [41]. We also acknowledge that assessing ITH from bulk genomic profiles is intrinsically difficult, where the different available tools can provide inconsistent measures [20]. Moreover, bulk sequencing of a single region is likely to provide underestimated measures of ITH when compared with multi-region or single-cell studies.”

2. The authors should mention and to discuss the following papers: (i) Lahoz S et al. *Journal of Pathology* 2022; 257: 68-81.; (ii) Chowdhury S et al, *Cancers* 2021; 13: 4923.; (iii) Banerjee S et al. *iScience* 2021; 24: 102718.

Response: We acknowledge the reviewer for these references and included the previous findings in our Discussion: Chowdhury et al 2021 (lines (466-471); Banerjee et al 2021 lines (492-494; 541-542); Lahoz et al 2022 (lines 535-538);

Rewritten texts:

“In fact, a recent study suggests that the CMS1 and CMS4 signal is mostly derived from immune and stromal cells, respectively [42], with little contribution from tumor epithelial cells. Besides the differential content in stromal and immune cells, CRC tumors also showed ITH at the level of epithelial cells. Indeed, a variety of CMS-like signatures were detected inside each tumor sample, as previously described [30,42]”

“A recent study using multi-region sequencing of 68 patients confirmed this distinction, suggesting a higher complexity of left-sided tumors [45].”

“A recent study with 84 untreated stage II CRC patients showed that a higher burden of CNAs, particularly subclonal, was associated with an increased proportion of recurrence [60], while the same was not found for SNVs.”

“Also, a single-cell genomic profile of primary-metastasis pairs of two CRC patients, revealed a late dissemination model with polyclonal seeding [61], with results from another study also suggesting that metastasizing tissues are often formed from multiple clones [45].”

Point-by Point response to Reviewer #2

Sobral et al. have presented a comprehensive analysis of the genetic and microenvironmental heterogeneity of colorectal cancer using a prospective biobank of primary and metastatic tumor tissues (including matched pairs), compared to publicly available data sets. Specifically, they highlight unique features of intra-tumoral heterogeneity that favor tumor progression and metastasis. The authors have revised the manuscript to outline the advantages of their findings more clearly over prior publications. They have made additional changes in response to prior reviewer comments that strengthen the overall work. There are a few minor edits that would improve the manuscript.

Response: We would like to thank the referee for her/his careful review of our manuscript and appreciate the positive feedback relative to the revised version.

1. Line 86 – remove “ed” on developed

Response: We acknowledge the reviewer for this note and corrected the mistake (line 82).

2. Lines 256-258 – is there an explanation for why KRAS mutations were not detected in CMS3 samples? Has this been observed in other studies?

Response: Although some CMS3 samples harbored KRAS mutations (33%), we could not observe an enrichment as reported in other studies. One possible explanation can be the small cohort size, especially for CMS3 subtype (12 samples). We could not find other studies with similar proportion of CMS3 mutated samples.

3. Line 387 – change to “CMS classification” rather than CRC classification

Response: We acknowledge the reviewer for this note and corrected the designation to CMS classification (line 231).

4. Line 658 – change to “A favorable outcome” rather than The favorable

Response: We acknowledge the reviewer for this note and corrected the mistake (line 501).

5. Line 677 – can the authors provide references that support the microenvironmental signatures (e.g., endothelial stalk cells, dendritic cells, fibroblasts) postulated as biomarkers for metastatic potential? There are several colorectal cancer spatial transcriptomic and single cell RNAseq experimental data sets published that could be useful to look at and cite (e.g., <https://doi.org/10.1016/j.ccell.2022.02.013>; <https://doi.org/10.1038/s41467-022-29366-6>).

Response: We acknowledge the reviewer for these references and included the previous works in the Discussion to support the microenvironmental signatures (lines 510-516; 518-520).

Rewritten text:

“Although, tumor-associated endothelial cells have usually been associated with the promotion of tumor growth and resistance to treatment [48], our results suggest that at least a subset of tumor-associated endothelial cells may have a protective role in preventing relapse. Similarly, a dual role has been also described for dendritic cells by promoting tumor invasiveness and worse outcome [49] or association with immune activation and clearance

[50]. Indeed, our analyses associate general markers of dendritic cells with less prevalence of relapse. On the other hand, enrichment of myofibroblasts and pro-inflammatory macrophages were shown to induce pro-tumorigenic and immunosuppressive mechanisms that lead to tumor progression and metastases [51]. In fact, the presence of myofibroblasts has been previously associated with tumor invasiveness in colorectal cancer [52–54].”

6. Line 696 – I would suggest making this a separate section “Limitations of Study” to acknowledge the small study sample size and include the limitations for assessing ITH using bulk sequencing analysis vs single cell analyses.

Response: We understand the reviewer’s comments and have rewritten the Discussion to refer all the limitations of our study in the same paragraph (lines 436-442).

Rewritten text:

“Nonetheless, we acknowledge limitations due to small sample size, particularly given the small number of metastasizing tumors which reduce statistical power. Nevertheless, we reinforced our findings using independent cohorts of early primary tumors from TCGA [3] and primary-metastasis pairs publicly available [41]. We also acknowledge that assessing ITH from bulk genomic profiles is intrinsically difficult, where the different available tools can provide inconsistent measures [20]. Moreover, bulk sequencing of a single region is likely to provide underestimated measures of ITH when compared with multi-region or single-cell studies.”